# Remote measurement based care (RMBC) interventions for mental health—Protocol of a systematic review and meta-analysis

Felix Machleid[1,2☯]*, Twyla Michnevich[1☯], Leu Huang[3], Louisa Schröder-Frerkes[4], Caspar Wiegmann[5,6], Toni Muffel[6], Jakob Kaminski[1,2,6]

1 Department of Psychiatry and Psychotherapy, Charité Campus Mitte, Charité – Universitätsmedizin Berlin, Corporate Member of Freie Universität Berlin, Humboldt-Universität zu Berlin, and Berlin Institute of Health, Berlin, Germany, 2 Berlin Institute of Health at Charité – Universitätsmedizin Berlin, Berlin, Germany, 3 Department of Infectious Diseases and Respiratory Medicine, Charité Campus Virchow-Klinikum, Charité – Universitätsmedizin Berlin, Berlin, Germany, 4 Clinic for Psychiatry, Psychotherapy and Psychosomatics, Krankenhaus am Urban, Berlin, Germany, 5 Clinics for Psychiatry and Psychotherapy, Clinics at the Theodor-Wenzel-Werk, Berlin, Germany, 6 Recovery Cat GmbH, Berlin, Germany

☯ These authors contributed equally to this work.
* felix.machleid@charite.de

**Data Availability Statement:** No datasets were generated or analysed during the current study. All

## Abstract

### Background

Poor management of mental illnesses is associated with lower treatment adherence, chronification, avoidable re-hospitalisations, and high costs. Remote measurement based care (RMBC) interventions have gained increasing relevance due to its potential in providing a comprehensive and patient-centric approach to mental health management.

### Objectives

The systematic review and meta-analysis aims to provide a comprehensive overview and analysis of existing evidence on the use of RMBC for patients with mental illness and to examine the effectiveness of RMBC interventions in alleviating disorder-specific symptoms, reducing relapse and improving recovery-oriented outcomes, global functioning, and quality of life.

### Methods and analysis

Our multidisciplinary research team will develop a comprehensive search strategy, adapted to each electronic database (PubMed, Medline, Embase, and PsychINFO) to be examined systematically. Studies with patients formally diagnosed by the International Classification of Diseases or the Diagnostic and Statistical Manual of Mental Disorders which include assessment of self-reported psychiatric symptoms will be included. Publications will be reviewed by teams of independent researchers. Quality of studies will be assessed using the Cochrane Collaboration's tool for assessing risk of bias. Outcomes cover symptom-focused or disease-specific outcomes, relapse, recovery-focused outcomes, global functioning, quality of life and acceptability of the intervention. Further data that will be extracted

relevant data from this study will be made available upon study completion.

**Funding:** Felix Machleid is funded by the Junior Digital Clinician Scientist Program of the Berlin Institute of Health. Jakob Kaminski was supported by the digital health accelerator that facilitated the spin-off process of Recovery Cat (a software company dedicated to digital health in psychiatry) from Charité. No additional specific grant from any funding agency in the public, commercial or not-for-profit sectors was obtained. The funders had no role in study design, data collection and analysis, decision to publish, or preparation of the manuscript.

**Competing interests:** Jakob Kaminski is shareholder and managing director of Recovery Cat GmbH. Toni Muffel is an employee at Recovery Cat GmbH, Caspar Wiegmann received honorary from Recovery Cat for consulting. This does not alter our adherence to PLOS ONE policies on sharing data and materials.

includes study characteristics, target population, intervention, and tracking characteristics. Data will be synthesised qualitatively, summarising findings of the systematic review. Randomised controlled trials (RCTs) will be considered for meta-analysis if data is found comparable in terms of mental illness, study design and outcomes. Cumulative evidence will be evaluated according to the Grading of Recommendations Assessment, Development and Evaluation framework.

## Trial registration

**Trial registration number**: PROSPERO CRD42022356176.

## Introduction

Mental illnesses are estimated to be one of the highest global burdens of disease [1]. Managing these illnesses presents challenges to patients and clinicians alike. Often, subjective symptom reporting [2,3], memory bias [4], complex treatment interactions, poorly coordinated transitions from inpatient to outpatient settings [5], and short, infrequent appointments in outpatient and private practices [6] result in the loss of information about symptom progression and treatment side effects. Poor continuity of care increases the risk of lower treatment adherence [7], worsening of disease [7,8], avoidable re-hospitalisations [5,8], and associated costs [8].

To address these challenges, there has been a surge in development of diagnostic and therapeutic mobile mental health (MMH) technologies. One of the leading use cases for MMH is remote measurement based care (RMBC), which involves asynchronous assessment of (electronic) patient reported outcomes (ePROs) outside of clinical encounters and their use for clinical decision-making and scheduling [9,10]. In addition to 'traditional' retrospective PRO assessment formats (e.g. validated questionnaires), ambulatory and diary methods have gained interest, as they offer unique insights into patients' experience of symptoms and well-being in their natural surroundings [11]. Such methods, termed ecological momentary assessment (EMA), have evolved with technological advances that allow self-reporting of symptoms in an internet or mobile context, including web/online, text messaging or phone-call based methods [12].

RMBC interventions have gained significance in the context of mental health management because of their ability to continuously track PROs. They offer the potential to improve the detection of deterioration, to personalise treatment plans, and improve accessibility to high quality mental health services [9].

As technology rapidly advances, studies have shown strong evidence in favour of RMBC, which can improve clinical outcomes and increase treatment adherence [13–15]. For example, a study including 6,424 participants with various psychiatric diagnoses found that continuous feedback to therapists on the course of symptoms was associated with a doubling of therapeutic effects related to individual (or symptomatic) functioning, interpersonal relationships, and social role performance [16].

Moreover, the benefits of measurement-based care (MBC) during in-person clinical encounters have been recognised before the rise of RMBC. MBC has been linked to faster remission [17,18] compared to treatment as usual (TAU) and fewer missed outpatient appointments [19,20]. MBC can also support clinicians in adjusting treatment quickly and effectively [10,21]. Patients consider MBC helpful [22], and it may improve doctor patient communication or increase treatment motivation [23,24].

Despite these advantages, asynchronous MBC using digital solutions is not yet widespread in clinical practice resulting in various implementation efforts [25]. Although there is limited robust scientific evidence from randomised controlled trials or longitudinal studies [9], there is a significant number of pilot and feasibility studies on RMBC systems. However these studies often have the typical limitations of academic research such as insufficient power or limited bias reduction strategies [26,27]. This leads to heterogeneity in available data and necessitates regular systematic evaluations identifying general trends or effects to increase the adoption of MBC and MMH in clinical practice.

In a 2018 systematic review by Goldberg et al. containing 13 RCTs on RMBC the results were promising [9]. The review highlighted the short-term feasibility and acceptability of RMBC. However, only three studies isolated the effects of RMBC experimentally, with one reporting greater symptom improvement in the RMBC group and two finding no differences between intervention group and controls. In the planned systematic review and meta-analysis, we aim to provide a comprehensive overview of the current evidence on RMBC in psychiatric care and update the above mentioned findings by Goldberg and colleagues [9]. Specifically, we will focus on interventions that aim to alleviate disorder-specific symptoms, reduce relapse, and improve recovery-oriented outcomes, global functioning, and quality of life. As the data allows, we will also conduct meta-analyses of relevant outcomes.

## Methods

### Protocol and registration

This protocol follows the guidelines of PRISMA-P (preferred reporting items for systematic review and explanation meta-analysis protocols) checklist (S1 Table, [8]). The systematic review and meta-analysis were registered at PROSPERO (reference CRD42022356176).

### Eligibility criteria

The research question and inclusion and exclusion criteria (S2 Table) were formulated using the PICOS (population, intervention, comparison, outcome, study) framework [28].

### Population

The review will include studies published before August 24th, 2022 examining adults ($\geq$18 years) with mental health disorders according to the International Statistical Classification of Diseases and Related Health Problems (ICD) and/ or the Diagnostic and Statistical Manual (DSM). Interventions delivered solely to family members (either targets of the intervention or in addition to patients) will be excluded.

### Interventions

We will include interventions that allow the assessment of self-reported symptoms of mental illness and other aspects of well-being. This feature is not required to be the predominant element of the intervention. Thus, studies that include an evidence-based form of therapy (e.g. cognitive-behavioural therapy, psychodynamic therapies, behaviour therapy or behaviour modification, etc.) plus an RMBC element will also be included.

### Comparison

Due to the variety of included study designs, no specific comparison to a control group is intended.

## Outcomes

Studies must report quantitative data, such as changes in symptom-related outcomes, recovery-related outcomes (e.g. empowerment, self-efficacy, hope, social connectedness), (global) level of functioning, relapse, or quality of life. Symptom tracking entirely derived from passive sensing or passive monitoring, as well as quantitative data in the form of correlative data analyses and statistical prediction models, will be excluded.

## Study designs

We will comprehensively consider randomised studies, including RCTs, cluster RCTs or factorial RCTs, non-randomised studies, including observational, cohort, cross-sectional or case-control studies; mixed methods studies, and feasibility or pilot studies with available full texts written in English or German.

## Information sources and search strategy

The multidisciplinary research team developed a comprehensive search strategy by screening reference lists of scientific and grey literature, including MeSH terms related to (1) mental disorders and psychological distress, (2) measurement-based care, and (3) digital technologies. The syntax of each database search will be slightly modified due to database specifications and the searches will be conducted using PubMed, Medline, Embase, and PsychINFO. The search strategy for each database can be found in S3 Table.

## Study selection

For all reports included in the synthesis, data will be collected, extracted, and reviewed in a Google Sheets spreadsheet developed by the research team. In an initial screening, titles and abstracts will each be evaluated in batches by three subgroups of independent researchers (TwM&LS, FM&LH, JK&CW&ToM). In a second screening step, full-text articles of the selected records will be retrieved and imported to the Zotero reference management software. The subgroups will assess a different batch of full-text articles than the ones they formerly screened. Disagreements between the researchers will be discussed to reach a consensus. When no consensus can be reached within the pair/trio, the discussion will be conducted within the entire research team. The rationale for the exclusion of each full text will be provided.

## Data extraction

We plan to extract various data: study identification (authors, year of publication, doi, URL), population (e.g. number of cases and controls, diagnosis, age, gender, years pre-university education), tracking- (e.g. mode, items, frequency) and study characteristics (e.g. design, hypotheses, study site, duration, randomisation, post-assessment period, follow-up, outcomes, response rate). Outcomes will be categorised into six categories: (1) symptom-focused or disease-specific outcomes, (2) relapse, (3) recovery-focused outcomes, (4) (global) functioning, (5) quality of life and (6) acceptability. We will consider metrics and timing of measurements for each outcome measurement tool and instrument. We will code p-values and standardised effect sizes (e.g. Cohen's d, hazard ratio, odds ratio) when available. One researcher will extract all study features, and an additional researcher will review the coding. Discrepancies will be resolved through discussions with a third researcher.

## Assessment of bias

Risk of bias in randomised studies will be examined using the Cochrane Collaboration's tool version 2.0 [29]. We will use Funnel plot methods to examine publication bias in analyses with more than 10 studies included [30].

## Data synthesis

For all studies included, a detailed description of the data items extracted and relevant results will be provided in narrative synthesis and respective tables.

## Meta-analysis

Data analysis will be conducted using RStudio [31]. All types of randomised clinical studies will be considered for meta-analysis. To account for potential variations in true effects in the studies due to differences in study populations, interventions, and target behaviours, all meta-analyses will be conducted as a random-effects analyses [32].

Different diagnostic groups such as psychosis, depression or mania will be examined separately. If k>2 studies report the same outcome, they will be meta-analytically pooled. When a single study uses multiple measures to report the same outcome, the one defined as the primary outcome of the study will be favoured, or the more common measure in other studies will be chosen. Same outcomes will be examined across all diagnostic groups, with the inclusion of the diagnosis factor as a covariate in statistical analysis. When different measures are used across studies for similar outcomes, effect sizes will be standardised for pooling [32]. We will analyse studies that include treatment as usual and active control groups together. If reported, we will use intention to treat data for our analyses.

We will express effect sizes for continuous measures as standardised mean differences (SMDs) and their 95% confidence intervals calculated using the pooled standard deviation of the interventions. SMDs will be presented as values of Hedges' *g*. For dichotomous measures, we will calculate risk ratios and combine the studies using the Mantel-Haenszel method. The number needed to treat (NNT) will be calculated to further illustrate the clinical relevance of RMBC interventions.

In the case of cluster RCTs, we consider each cluster as a distinct entity, using summary measures from each individual cluster for meta-analysis. In instances where the RCT does not provide adequate details for analysis, we rely on cluster-specific information, such as the intra-class correlation coefficient, to perform an approximate analysis [30]. In factorial RCTs, data from each treatment arm is extracted separately and treated as an individual study when relevant to meta-analysis. All findings will be reported transparently [30].

Sensitivity analyses will be carried out to examine the effect of a specific study on the pooled outcomes. Subgroup analyses separating population characteristics (e.g. mean age, gender, years of pre-university education, severity of illness), study characteristics or RMBC intervention type will be conducted to reduce heterogeneity of pooled estimates.

## Heterogeneity

We will assess heterogeneity using the $I^2$ statistic, *p*-value of $\chi^2$ test and visual inspection of forest plots.

## Confidence in cumulative evidence

Cumulative evidence will be evaluated according to the Grading of Recommendations Assessment, Development and Evaluation (GRADE) framework.

## Dissemination

We will publish the results of our study in a peer-reviewed journal in the field of psychiatry, digital health or telemedicine, and present them at international conferences and workshops.

## Conclusion

Our study will summarise existing evidence on the use of RMBC interventions in mental health care and offer insights into their impact on clinical-, health service utilisation-, recovery- and quality of life related outcomes.

## Supporting information

**S1 Table. PRISMA-P 2015 checklist.**
(DOCX)

**S2 Table. Inclusion and exclusion criteria related to the PICOS design.**
(DOCX)

**S3 Table. Search syntax, date of searches, and number of results returned for each database, number of references for review.**
(DOCX)

## Acknowledgments

We want to thank our colleagues at Recovery Cat for their support and thoughtful input.

## Author Contributions

**Conceptualization:** Felix Machleid, Twyla Michnevich, Leu Huang, Louisa Schröder-Frerkes, Caspar Wiegmann, Toni Muffel, Jakob Kaminski.

**Methodology:** Felix Machleid, Twyla Michnevich.

**Writing – original draft:** Felix Machleid, Twyla Michnevich, Toni Muffel.

**Writing – review & editing:** Felix Machleid, Twyla Michnevich, Leu Huang, Louisa Schröder-Frerkes, Caspar Wiegmann, Jakob Kaminski.

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
