## [Decision Letter · Decision Letter 0]

30 Mar 2023

PONE-D-23-00480Remote measurement based care (RMBC) interventions for mental health - protocol of a systematic review and meta-analysis.PLOS ONE

Dear Dr. Machleid,

Thank you for submitting your manuscript to PLOS ONE. After careful consideration, we feel that it has merit but does not fully meet PLOS ONE’s publication criteria as it currently stands. Therefore, we invite you to submit a revised version of the manuscript that addresses the points raised during the review process.

ACADEMIC EDITOR: Dear authors, your paper has been positively evaluated, therefore it could be accepted in our journal with minor amendments. Notably, one of the reviewer suggested to revise the manuscript, recommending language editing, which seems a valuable advise to improve paper's readability.

We look forward to receiving your revised manuscript.

Kind regards,

Jacopo Demurtas

Academic Editor

PLOS ONE

“JK is shareholder and managing director of Recovery Cat GmbH. ToM is an employee at Recovery Cat GmbH, CW received honorary from Recovery Cat for consulting.  “

Reviewers' comments:

Reviewer's Responses to Questions

**Comments to the Author**

1. Does the manuscript provide a valid rationale for the proposed study, with clearly identified and justified research questions?

Reviewer #1: Yes

Reviewer #2: Yes

2. Is the protocol technically sound and planned in a manner that will lead to a meaningful outcome and allow testing the stated hypotheses?

Reviewer #1: Yes

Reviewer #2: Yes

3. Is the methodology feasible and described in sufficient detail to allow the work to be replicable?

Reviewer #1: Yes

Reviewer #2: Yes

4. Have the authors described where all data underlying the findings will be made available when the study is complete?

Reviewer #1: Yes

Reviewer #2: Yes

5. Is the manuscript presented in an intelligible fashion and written in standard English?

Reviewer #1: Yes

Reviewer #2: Yes

6. Review Comments to the Author

You may also provide optional suggestions and comments to authors that they might find helpful in planning their study.

Reviewer #1: The manuscript is original and of interest for journal audience.

Language editing highly recommended

Reviewer #2: Due to the scientific importance of the subject matter and the appropriateness of the approach and methodological and statistical correctness, I think it is appropriate that it be accepted.

7. PLOS authors have the option to publish the peer review history of their article (what does this mean?). If published, this will include your full peer review and any attached files.

Reviewer #1: No

Reviewer #2: No

---

## [Author Response · Author response to Decision Letter 0]

8 Apr 2023

Rebuttal Letter 

Dear Mr. Demurtas,

Thank you for taking the time to review our protocol titled Remote measurement based care (RMBC) interventions for mental health - Protocol of a systematic review and meta-analysis. We appreciate the effort the editorial team and the reviewers have put into providing feedback on our work. We have taken your comments into careful consideration and have made the necessary revisions: 

● According to one of the reviewers' comments, we have reworked the language of our protocol and improved its readability. 

● Furthermore we have edited the manuscript so it meets PLOS ONE's style requirements, updated the financial disclosure and competing interests section. 

Thank you once again for your valuable feedback and I look forward to hearing back from you.

Sincerely,

Dr. Felix Machleid

---

## [Decision Letter · Decision Letter 1]

28 Jul 2023

PONE-D-23-00480R1Remote measurement based care (RMBC) interventions for mental health - protocol of a systematic review and meta-analysis.PLOS ONE

Dear Dr. Machleid,

Thank you for submitting your manuscript to PLOS ONE. After careful consideration, we feel that it has merit but does not fully meet PLOS ONE’s publication criteria as it currently stands. Therefore, we invite you to submit a revised version of the manuscript that addresses the points raised during the review process.

ACADEMIC EDITOR: Congratulation for the revised manuscript. I would like to suggest you to address the few issues raised by the reviewer regarding English editing. 

 We look forward to receiving your revised manuscript.

Kind regards,

Dirceu Henrique Paulo Mabunda, M.D.

Academic Editor

PLOS ONE

Journal Requirements:

Reviewers' comments:

Reviewer's Responses to Questions

**Comments to the Author**

1. Does the manuscript provide a valid rationale for the proposed study, with clearly identified and justified research questions?

Reviewer #3: Yes

Reviewer #4: Yes

2. Is the protocol technically sound and planned in a manner that will lead to a meaningful outcome and allow testing the stated hypotheses?

Reviewer #3: Partly

Reviewer #4: Yes

3. Is the methodology feasible and described in sufficient detail to allow the work to be replicable?

Reviewer #3: Yes

Reviewer #4: Yes

4. Have the authors described where all data underlying the findings will be made available when the study is complete?

Reviewer #3: Yes

Reviewer #4: Yes

5. Is the manuscript presented in an intelligible fashion and written in standard English?

Reviewer #3: Yes

Reviewer #4: Yes

6. Review Comments to the Author

You may also provide optional suggestions and comments to authors that they might find helpful in planning their study.

Reviewer #3: This article presents a review protocol focusing on the use of remote measurement-based care interventions for mental health. The article addresses the growing relevance of RMBC interventions in evaluating treatment effects and closing gaps in existing care pathways by capturing patient-reported outcome measures. I received a revised version of the manuscript that is already sufficiently good. Indeed, I provide a minor revision, considering the point I suggest to incorporate that are summarized below are small refinements:

# Abstract

- Better clarify the significance of RMBC interventions. The sentence "by providing a low-threshold means of capturing patient reported outcome measures" is unclear (page 10).

- Use the singular verb form 'aims' to highlight the collective action of the systematic review and meta-analysis (page 10).

- Specify whether the review includes studies other than randomized controlled trials (RCTs) and provide more specific details (page 10).

# Introduction

- Clarify whether MBC or RMBC is the correct abbreviation. The authors seems to use them as interchangeable terms (page 12).

- Address the lack of references and supporting evidence for the last paragraph of page 12 ("MMH tech companies develop comprehensive solutions, yet their scientific evaluation is often less granular than that in a university setting, lacking scientific evidence. This constitutes a major barrier to dissemination for (mental) healthcare providers and health insurance companies. On the other hand, the surge of MMH technologies in the past years has led to a plethora of pilot and feasibility studies on RMBC systems developed and evaluated by clinical research teams, which have the typical limitations of academic research (e.g., insufficient power, limited bias reduction strategies) resulting in scattered evidence. This heterogeneity in available data necessitates regular systematic evaluations identifying general trends or effects to increase the adoption of MBC and MMH in clinical practice").

- Provide details on how the different results between diagnoses will be handled, in particular it should be done in methods section (page 13).

# Methods

## Protocol, registration and ethics

- Correct the doubled sentence "Since we will not collect any primary d ata of individualsSince no primary data of individuals will be collected," (page 13).

## Interventions

- I would suggest to rephrase "must not be the predominant" with something like “is not requested to be the predominant” (page 14).

- Correct the double comma in the sentence related to therapy types ("cognitive-behavioural therapy,, psychodynamic therapies") (page 14).

## Study designs

- Report the study designs of the studies to be included in detail, including appropriate nomenclature for non-randomized clinical studies (page 15).

## Information sources and search strategy

- Explain how the search was conducted separately, considering that PubMed is based on MEDLINE. Please, address this also in the abstract (page 15).

## Information sources and search strategy

- Clarify the sentence "The groups will assess a different batch of full-text articles than formerly screened, following the same decision making process described above." (page 16).

## Data extraction

- Specify in the "Data extraction" that you will extract the diagnosis and the phase of the disease, in addition to other data (page 16).

- Clarify the timepoints you will extract for the selected outcomes (page 16).

## Assessment of bias

- Provide clarification on the quality assessment instrument for non-randomized studies, and clarify why you decided to use the original Cochrane Collaboration's tool for risk of bias tool and not the newer 2.0 version (page 17).

## Meta-analysis

- Indicate the specific R packages to be used in the data analysis (page 17).

- Explain the approach for managing the cluster effect on outcomes measures and how you plan to combine study arms of factorial RCTs (page 17).

- Address whether the same outcome will be meta-analyzed across different diagnoses or separately (page 17).

Reviewer #4: Thank you for the opportunity to review this protocol for a systematic review and meta-analysis. I noted that this is Revision 1, although this is the first time, I have been asked to review it.

It appears the authors were asked to revise the readability of the text include the Tables as Supplementary material. Both of these points have been addressed and now the protocol now clearly written. As the growing burden of mental health problems is a global issue. I believe the paper should be accepted with minor changes.

I suggest the following minor corrections.

1. Methods: ‘Since we will not collect any primary data of individualsSince no

primary data of individuals will be collected.’ Please remove duplicate text.

2. Population: I suggest updating the inclusion criteria from August 2022 to 2023.

3. Study design. Please remove the full stop at end of the sentence.

4. Data extraction: It is not clear what is meant by ‘We will code p values and standardised effect sizes’. Please give an example.

5. Meta analysis: The following text should be in future text ‘all meta-analyses were conducted as a random-effects analyses’ in line with the rest of the section.

7. PLOS authors have the option to publish the peer review history of their article (what does this mean?). If published, this will include your full peer review and any attached files.

Reviewer #3: **Yes: **Alessandro Rodolico

Reviewer #4: No

---

## [Author Response · Author response to Decision Letter 1]

13 Oct 2023

Response to Reviewers 

Brief summary of changes: 

Reviewer 3: Thank you for your valuable feedback. We have addressed your suggestions as follows: We have clarified the significance of RMBC interventions, made distinctions between MBC and RMBC and added references of feasibility and pilot studies. Further, we have specified the search strategy, the study designs to be included, extraction details and included a paragraph on how we will manage cluster and factorial RCTs.

Reviewer 4: Thank you for your positive feedback and valuable suggestions. We have made the following minor corrections: We have clarified the meaning of "coding p-values and standardized effect sizes" and provided an example. Further, we have decided to maintain the inclusion criteria up to August 2022, as originally stated.

Reviewer #3: This article presents a review protocol focusing on the use of remote measurement-based care interventions for mental health. The article addresses the growing relevance of RMBC interventions in evaluating treatment effects and closing gaps in existing care pathways by capturing patient-reported outcome measures. I received a revised version of the manuscript that is already sufficiently good. Indeed, I provide a minor revision, considering the point I suggest to incorporate that are summarized below are small refinements:

# Abstract

R3: Better clarify the significance of RMBC interventions. The sentence "by providing a low-threshold means of capturing patient reported outcome measures" is unclear (page 10).

- Answer: The abstract has been rewritten and the phrase has been reworded in the introduction: 

- Edit: “Remote measurement based care (RMBC) interventions have gained increasing relevance by providing a comprehensive and patient-centered approach to mental health management.”

R3: Use the singular verb form 'aims' to highlight the collective action of the systematic review and meta-analysis (page 10).

- Answer: We have changed this to the singular form as per your suggestion.

R3: Specify whether the review includes studies other than randomized controlled trials (RCTs) and provide more specific details (page 10).

- Answer: We have added a sentence outlining the main inclusion criteria:

- Edit: “Studies with patients formally diagnosed by the International Classification of Diseases or the Diagnostic and Statistical Manual of Mental Disorders which include assessment of self-reported psychiatric symptoms will be included.”

# Introduction

R3: Clarify whether MBC or RMBC is the correct abbreviation. The authors seems to use them as interchangeable terms (page 12).

- Answer: The introduction has been clarified to distinguish between the broader concept of MBC and the specific application of RMBC. Measurement-Based Care (MBC) is an umbrella term that encompasses both traditional in-person assessment and Remote Measurement-Based Care (RMBC). While MBC also involves tracking patient-reported outcomes (PROs) during clinical encounters, RMBC specifically refers to the remote tracking of electronic PROs outside of traditional face-to-face appointments. In one instance there was an error and we have replaced “MBC” with “RMBC”: 

- Edits: “Overall, the results were promising, although only three studies isolated the effects of RMBC experimentally, with one reporting greater symptom improvement in the RMBC group and two finding no differences between intervention group and controls.”

R3: Address the lack of references and supporting evidence for the last paragraph of page 12 ("MMH tech companies develop comprehensive solutions, yet their scientific evaluation is often less granular than that in a university setting, lacking scientific evidence. This constitutes a major barrier to dissemination for (mental) healthcare providers and health insurance companies. On the other hand, the surge of MMH technologies in the past years has led to a plethora of pilot and feasibility studies on RMBC systems developed and evaluated by clinical research teams, which have the typical limitations of academic research (e.g., insufficient power, limited bias reduction strategies) resulting in scattered evidence. This heterogeneity in available data necessitates regular systematic evaluations identifying general trends or effects to increase the adoption of MBC and MMH in clinical practice").

- Answer: The part was rewritten and ambiguous passages with lack of evidence removed, references of feasibility and pilot studies were added. 

- Edit: “Despite these advantages, asynchronous MBC using digital solutions is not yet widespread in clinical practice resulting in various implementation efforts [25]. Although there is limited robust scientific evidence from randomised controlled trials or longitudinal studies [9], there is a significant number of pilot and feasibility studies on RMBC systems. However these studies often have the typical limitations of academic research such as insufficient power or limited bias reduction strategies [26,27]. This leads to heterogeneity in available data and necessitates regular systematic evaluations identifying general trends or effects to increase the adoption of MBC and MMH in clinical practice.” 

R3: Provide details on how the different results between diagnoses will be handled, in particular it should be done in methods section (page 13).

- Answer: We have added a passage explaining that different diagnoses such as psychosis, depression or mania will be examined separately.

- Edit: “Different diagnostic groups such as psychosis, depression or mania will be examined separately.”

# Methods

## Protocol, registration and ethics

R3: Correct the doubled sentence "Since we will not collect any primary data of individualsSince no primary data of individuals will be collected," (page 13).

- Answer: Please excuse the oversight, we have deleted the duplicate phrase.

## Interventions

R3: I would suggest to rephrase "must not be the predominant" with something like “is not requested to be the predominant” (page 14).

- Answer: We have rephrased to “is not required to be the predominant”.

R3: Correct the double comma in the sentence related to therapy types ("cognitive-behavioural therapy,, psychodynamic therapies") (page 14).

- We have deleted the extra comma.

## Study designs

R3: Report the study designs of the studies to be included in detail, including appropriate nomenclature for non-randomized clinical studies (page 15).

- Answer: An update of the study selection was given with a specification of the non-radnomised studies included. 

- Edit: “We will comprehensively consider randomised studies, including randomised controlled trials (RCTs), cluster RCTs or factorial RCTs, non-randomised studies, including observational longitudinal, cohort, cross-sectional or case-control studies; mixed methods studies, and feasibility or pilot studies with available full texts written in English or German. 

## Information sources and search strategy

R3: Explain how the search was conducted separately, considering that PubMed is based on MEDLINE. Please, address this also in the abstract (page 15).

- Answer: The PubMed search will still be conducted. A comprehensive search strategy will be developed and its syntax adapted for each database to be systematically assessed. PubMed goes beyond the scope of Medline and is a vital source of peer-reviewed literature. By including both we ensure covering an extensive pool of literature and maximize the scope and depth of the review.

## Information sources and search strategy

R3: Clarify the sentence "The groups will assess a different batch of full-text articles than formerly screened, following the same decision making process described above." (page 16).

- Answer: We have reorganized and slightly reworded the entire paragraph to make the screening process more clear.

- Edit: “For all reports included in the synthesis, data will be collected, extracted and reviewed in a Google Sheets spreadsheet developed by the research team. In an initial screening, titles and abstracts will each be evaluated in batches by three subgroups of independent researchers (TwM&LS, FM&LH, JK&CW&ToM). In a second screening step, full-text articles of the selected records will be retrieved and imported to the Zotero reference management software. The subgroups will assess a different batch of full-text articles than the ones they formerly screened. Disagreements between the researchers will be discussed to reach a consensus. When no consensus can be reached within the pair/trio, the discussion will be conducted within the entire research team. The rationale for the exclusion of each full text will be provided.” 

## Data extraction

R3: Specify in the "Data extraction" that you will extract the diagnosis and the phase of the disease, in addition to other data (page 16).

- Answer: Sorry for this oversight, we have added the diagnosis to the data extraction section. We do not plan to extract the phase of the disease, in the sense of e.g. acute, sub-acute. 

- Edit: “population (e.g. number of cases and controls, diagnosis, age, gender, years pre-university education)”

R3: Clarify the timepoints you will extract for the selected outcomes (page 16).

- Answer: We will extract data from all time points (e.g. baseline and follow-up timpoints) defined in the study protocol, and have added this explanation to the data extraction section: 

- Edit: Outcomes from all time points (e.g. baseline and follow-ups) defined in the study protocol will be extracted and will be categorised as follows into six categories: (1) symptom-focused or disease-specific outcomes, …”. 

## Assessment of bias

R3: Provide clarification on the quality assessment instrument for non-randomized studies, and clarify why you decided to use the original Cochrane Collaboration's tool for risk of bias tool and not the newer 2.0 version (page 17).

- Answer: Thank you for this suggestion, we have included version 2.0 into our analysis. Non-randomised studies will not be assessed for risk of bias as they will not be used for the meta-analysis. 

- 

## Meta-analysis

R3: Indicate the specific R packages to be used in the data analysis (page 17).

- Answer: We have not decided on the packages yet. 

R3: Explain the approach for managing the cluster effect on outcomes measures and how you plan to combine study arms of factorial RCTs (page 17).

- Answer: We included a paragraph on how to we’ll manage cluster and factorial RCTs

- Edit: “In the case of cluster RCTs, we consider each cluster as a distinct entity, using summary measures from each individual cluster for meta-analysis. In instances where the RCT does not provide adequate details for analysis, we rely on cluster-specific information, such as the intraclass correlation coefficient, to perform an approximate analysis [30]. In factorial RCTs, data from each treatment arm is extracted separately and treated as an individual study when relevant to meta-analysis. All findings will be reported transparently [30].” 

R3: Address whether the same outcome will be meta-analyzed across different diagnoses or separately (page 17).

- Answer / Edit: “Same outcomes will be examined across all diagnostic groups, with the inclusion of the diagnosis factor as a covariate in statistical analysis.”

Reviewer #4: Thank you for the opportunity to review this protocol for a systematic review and meta-analysis. I noted that this is Revision 1, although this is the first time, I have been asked to review it.

It appears the authors were asked to revise the readability of the text include the Tables as Supplementary material. Both of these points have been addressed and now the protocol now clearly written. As the growing burden of mental health problems is a global issue. I believe the paper should be accepted with minor changes.

I suggest the following minor corrections.

R4: 1. Methods: ‘Since we will not collect any primary data of individuals since no primary data of individuals will be collected.’ Please remove duplicate text.

- Answer: Please excuse the oversight which was also noted by Reviewer 1, we have deleted the duplicate phrase.

R4: 2. Population: I suggest updating the inclusion criteria from August 2022 to 2023.

- Answer: Thank you for this suggestion, however we decided to focus on the studies published until August 2022. 

R4: 3. Study design. Please remove the full stop at end of the sentence. 

- Answer: We have removed the full stop. 

R4: 4. Data extraction: It is not clear what is meant by ‘We will code p values and standardised effect sizes’. Please give an example.

- Answer / Edit: We have clarified the sentence: “For relevant outcomes, we will extract measures of statistical significance (p-values) We will code p-values and standardised effect sizes (e.g. Cohen’s d, hazard ratio, odds ratio) when available. “

R4: 5. Meta analysis: The following text should be in future text ‘all meta-analyses were conducted as a random-effects analyses’ in line with the rest of the section.

- Answer: Sorry for the oversight, we have corrected the sentence to future tense.

---

## [Decision Letter · Decision Letter 2]

16 Jan 2024

Remote-measurement based care (RMBC) interventions for mental health - protocol of a systematic review and meta-analysis.

PONE-D-23-00480R2

Dear Dr. Machleid,

We’re pleased to inform you that your manuscript has been judged scientifically suitable for publication and will be formally accepted for publication once it meets all outstanding technical requirements.

Kind regards,

Qin Xiang Ng, MBBS, GDMH, MPH

Academic Editor

PLOS ONE

Additional Editor Comments (optional):

Reviewers' comments:

Reviewer's Responses to Questions

**Comments to the Author**

1. Does the manuscript provide a valid rationale for the proposed study, with clearly identified and justified research questions?

Reviewer #5: Yes

2. Is the protocol technically sound and planned in a manner that will lead to a meaningful outcome and allow testing the stated hypotheses?

Reviewer #5: Yes

3. Is the methodology feasible and described in sufficient detail to allow the work to be replicable?

Reviewer #5: Yes

4. Have the authors described where all data underlying the findings will be made available when the study is complete?

Reviewer #5: Yes

5. Is the manuscript presented in an intelligible fashion and written in standard English?

Reviewer #5: Yes

6. Review Comments to the Author

You may also provide optional suggestions and comments to authors that they might find helpful in planning their study.

Reviewer #5: This protocol has been modified, and I think the author's revisions meet the requirements for publication.

7. PLOS authors have the option to publish the peer review history of their article (what does this mean?). If published, this will include your full peer review and any attached files.

Reviewer #5: No

---

## [Editor Report · Acceptance letter]

9 Feb 2024

PONE-D-23-00480R2 

PLOS ONE

Dear Dr. Machleid, 

I'm pleased to inform you that your manuscript has been deemed suitable for publication in PLOS ONE. Congratulations! Your manuscript is now being handed over to our production team.

Kind regards, 

on behalf of

Dr. Qin Xiang Ng 

Academic Editor

PLOS ONE